# Suicide on YouTube:Factors engaging viewers to a selection of suicide-themed videos

**Eun Ji Jung[1,2], Seongcheol Kim[2]***

1 Smart Study Co., Ltd, Seoul, Republic of Korea, 2 School of Media and Communication, Korea University, Seoul, Republic of Korea

* hiddentrees@korea.ac.kr

## Abstract

Visual social media platforms can function as both facilitators and intervenors of concerning behaviors. This study focused on one of the health concerns worldwide, a leading cause of death related to mental health—suicide—in the context of a dominant visual social media platform, YouTube. This study employed content analysis method to identify the factors predicting viewer responses to suicide-themed content from the perspectives of *who'*s, *what'*s, and *how'*s of suicide-themed videos. The results of the hierarchical multiple regression showed that the characteristics of content provider and content expression were more significant predictors of viewer engagement than were the characteristics of the message. These findings have implications for not only platform service providers but also diverse groups of individuals who participate in online discussions on suicide. YouTube has the potential to function as a locus for open discussion, education, collective coping, and even the diagnosis of suicidal ideation.

## 1. Introduction

According to the latest data on suicide by the World Health Organization [1], nearly 800,000 people die every year due to suicide, meaning one person dies every 90 seconds. Suicide can occur at any time in life and is the second leading cause of death among individuals aged 15–29 years.

The role of the Internet, particularly social networking services (SNSs), on suicide-related thoughts and behaviors has been a topic of growing interest and debate. There have been long-standing concerns over how social networking services manage content that may negatively affect the psychological well-being of its audience, especially the young users. This became an urgent issue following the death of a British girl, Molly Russel, whose father, Ian Russel, stated in an interview with BBC that Instagram encouraged his daughter to commit suicide [2]. In 2017, Molly Russel, who was known to have been posting and searching for keywords related to suicide and self-harm, such as "cutting," "biting," and "burning," ended her own life. The posts that she "liked" were identified to be images that glorified suicide. This prompted the discussion on the need for an advanced platform policy to prevent such incidents from happening again.

Furthermore, there have been several incidents of self-expressive YouTubers ending their own lives. Jamey Rodemeyer, a 14-year-old YouTuber who actively expressed himself through

**Data Availability Statement:** All relevant data are within the manuscript and we provide our codebook.

**Funding:** This work was supported by the Ministry of Education of the Republic of Korea and the

National Research Foundation of Korea (NRF-2019S1A3A2099973) and the MSIT(Ministry of Science and ICT), Korea, under the ITRC (Information Technology Research Center) support program(IITP-2020-0-01749) supervised by the IITP(Institute of Information & Communications Technology Planning & Evaluation).

**Competing interests:** The authors have declared that no competing interests exist.

videos on matters of his sexuality; homophobia; and lesbian, gay, bisexual, and transgender rights, ended his life on 18 September 2011. Although the school counselors had advised him not to use social media to talk about his sexuality, he voiced his thoughts through his YouTube posts. He appeared to be strong as he shared videos about the "It Gets Better" project, which aimed to address prevention of teen suicide. His suicide was attributed to excessive hostile comments. This case shows that social media sites are becoming venues to share personal opinions and to express oneself—even painful thoughts [3]—but at the same time, are making it easier for cyber-bullies to target their victims [4].

The influence of social media on concerning behaviors is not limited to children and teen-agers alone. The debate lies in how media function—whether as a facilitator or as an intervenor of such behaviors. Considering the debate, this study aims to examine how deliverers of sui-cide-themed contents discuss suicide and to examine what factors, among content provider characteristics, story characteristics, and content expression characteristics, predict viewer engagement. The current study focused on one of the mainstream online video platforms, YouTube, as a site of analysis. It is not only a visual media platform but also a social network-ing service, which makes the investigation into the ongoing suicide-themed discussions on the platform worthwhile.

## 2. Literature review and research questions

### Suicide and the self

Suicide is defined as a "conscious act of self-induced annihilation" [5: p. 203] in the current Western society. A review of comparable concepts suggests that society has historically con-demned the act of killing oneself. Synonyms of suicide, such as self-killing, self-disembodiment, and self-murder, have shared stigmatizing connotations. This is because individuals are the con-stituents of society, where the sanity of one represents the degree of social health and well-being of the society as a whole. Suicidal thoughts and behaviors have also been considered pathologi-cal in the context of religion or morality. For example, the Protestants attribute melancholic self-disintegration to the temptation of Satan or a diabolical entity, which is distinguished from the inner self [6]. The concept evolved in the eighteenth century, encompassing terms from vol-untary death to involuntary self-killing. Since the term "suicide" indicates a change in attitude, relatively decriminalizing the act and the individual [6], it is used in the following discussions.

As a multidimensional malaise [5], one suicidal event involves "biological, psychological, intrapsychic, logical, conscious, and unconscious, interpersonal, sociological, cultural, and philosophical or existential" elements [7: p. 221]. In the field of suicidology, the utility of sui-cide note has been acknowledged [8]. Suicide notes provide information closest to the suicidal mind, which comprises multidimensional thoughts of an individual.

Suicide pertains to not only the self but also society, as society is regarded as an aggregate of many selves. There have been longstanding concerns over the diffusive nature of suicide. The diffusion process involves successful or unsuccessful suicide attempts that lead to serious sui-cidal ideations among others, and some of those contemplators make successful or unsuccessful attempts [9,10]. The diffusion of suicide in relation to the influence of media was studied follow-ing the widespread imitation of Werther's suicide, as described in the novel *The Sorrows of the Young Werther* by Johann Wolfgang von Goethe [10]. The matter lies in determining whether and how the media augment or intervene in the diffusion of suicidal thoughts and behaviors.

### Influence of media on suicidal individuals

Studies of the potential influence of media-publicized suicide stories of actual suicide have yielded inconsistent findings [11]. The existence of both media contagion effect and

intervention effect on suicide has been observed [1]. Media contagion refers to an adverse effect of media, whereas media intervention refers to a positive function of media.

**Media contagion effect.** The relationship between social media and socially concerning behaviors is complex. Social media can be hazardous to the vulnerable, as some online communities advocate extreme beliefs and behaviors, such as anorexia, suicide, and deliberate amputation, which are otherwise considered socially unacceptable [12]. Online discussion forums and social media chatrooms may facilitate socially undesirable behaviors as a result of peer pressure [13].

Recent studies that aimed to replicate and extend Phillips' imitation theorem suggest that widely publicized suicide stories trigger copycat suicides [10,11]. News or television coverage of suicide stories may provide role models for individuals at risk, which is related to a social learning theory of deviant behavior [14]. From this perspective, publicized suicide stories may encourage suicides in the real world, which makes it imperative to develop guidelines on how to deliver suicide stories in order to promote safe media content.

**Media intervention effect.** While a large body of research supports the propagative effect of media on suicide, another vein of research suggests that media has preventive functions. The protective function of media is referred to as the Papageno effect [15]. It was named after a character in Mozart's opera, *The Magic Flute*. Papageno becomes suicidal upon the loss of his beloved Papagena; however, he refrains from committing suicide thanks to a hopeful song by three elves. Media intervention effect suggests that media has a preventive function through education or collective coping with adverse situations.

The effectiveness of media on health-promoting activities was highlighted when articles that cited stories of individuals who refrained from executing their suicidal plans and of those who instead positively coped with adverse circumstances were published [15,16]. SNSs can help create social connections among individuals with shared experiences, raise awareness about prevention programs and crisis hotlines, and provide access to other available resources [12].

The advancement of media has enabled speedy diffusion of information without boundaries. The current study aimed to analyze suicide-themed content in a dominant visual media service platform, considering its reach and potential influence on the users.

## Characteristics of YouTube as a dominant media platform

**Social Networking Services (SNS) and YouTube.** A large number of SNSs exists, each with different technological affordances. They provide an array of features including profile-generating, making friends, commenting, and private messaging. Although designed to be available to a wide range of audiences, much of the populations for each site are segmented upon homogenous interests and purposes [17,18].

YouTube is an example of a community website that reflects the evolution of the Web environment [19], which can be characterized as follows. Web 1.0 environment was based on one-way information consumption, whereas in Web 2.0 environment, individual users and the networks among them are given the power; users have richer and more complex experiences; content distribution is not limited to content creators alone; and the boundaries of the devices are blurred [20]. With higher bandwidth, faster and more interactive experiences have been realized, providing users with rich visual media content such as audio and video streaming. According to YouTube Press, over 1.9 billion logged-in users visit YouTube monthly, which account for nearly one-third of the Internet users worldwide. YouTube provides multi-lingual experiences with a total of 80 different languages, covering about 95% of the entire Internet population. As it features a variety of video contents, YouTube, as a media-sharing website

that has become an SNS, is drawing the attention of everyone, regardless of age, gender, race or ethnicity, occupation, etc [17].

Technological advancement that has enabled SNSs to disseminate high volumes and a diverse range of information at a rapid rate across online networks has not always been discussed from a positive perspective. Continuous efforts have been made by multiple SNSs to develop an auto-filtering system to screen for and remove unsafe content. Following the incident of the live streaming of the New Zealand terror attack by Branton Tarrant, Facebook officially announced that they would adopt artificial intelligence (AI) technology to automatically filter harmful information. Twitter announced its plan to filter hate speech and spam tweets, while Tumbler, an image-based microblogging SNS, censors pornographic or illegal adult content. All these measures have been enforced, considering the influence of such mainstream platforms with a large user base.

**YouTube as a visual media platform.** Video-sharing websites have been gaining popularity on the Internet since the launch of YouTube in 2005. Compared to other media platforms, the most-frequently and widely-visited visual social media platform is YouTube, where the posts includes some form of visual information. Videos related to suicide or self-harm are concerning, because it is possible that such behaviors might become normalized, reinforced, or disinhibited [21,22] when the message is presented with visual effects. Extant studies show that the inclusion of visual material in a message facilitates longer memory retention [23], more accurate comprehension of the message [24], greater likelihood of reacting to a call to action presented in the message [25], and an increase in online engagement [26].

YouTube's current policy on suicide and self-injury states, "Content that promotes self-harm or is intended to shock or disgust users is not allowed on YouTube. We do allow users to post content discussing their experiences with depression, self-harm, or other mental health issues" [27]. When users come across content where the deliverer "expresses suicidal thoughts or is engaging in self-harm," they are advised to contact local authorities and press the flag button, which brings the post to YouTube's immediate attention, according to Andrea Faville, a spokesperson for YouTube [28]. The stance of the platform is that "the users should not be afraid to speak openly about the topics of mental health or self-harm," and the platform provides community guidelines, according to which content "promoting or glorifying suicide, providing instructions on how to self-harm, graphic images of self-harm posted to shock or disgust viewers" is banned [27]. The platform applies the same policies across all products and features, such as video posts, content descriptions, comments, and live streams. In instances of violation, the content is removed, and the creator is sent an email if it is their first time violating the policy. If it is not the first time, the creator is given a strike; three strikes result in channel termination.

These measures taken by platform service providers can be understood from the perspectives of both the Werther effect and Papageno effect. Studies of the relationship between the media coverage of suicide and suicidal behaviors in the real world have yielded inconsistent findings [11].

## Characteristics of the deliverer

The purposes of watching health-related YouTube videos include social utility, convenient information-seeking, leisure, and entertainment [29]. With increasing popularity of health-related social media usage, it is important to pay attention to the characteristics of the deliverers of the content. Given that shared content on YouTube is a source of health information and reflects one's experiences and emotions, the credibility and the diversity of the source are pivotal concerns.

Founded on the user-generated content (UGC) model, the contents on YouTube are created by its own users [30]. Content creators on this platform are commonly referred to as "YouTubers." Those who satisfy the community's needs through their content gain popularity and become so-called "influencers," who are considered micro-celebrities. New media scholar David Marshall [31] identified a transition from "representational" to "presentational" media and culture. In the age of social media and self-created content, the public self, public-private self, and transgressive intimate self are presented. This develops into the establishment of trust, credibility, and a sense of closeness between the creators and audience. The boundary between legacy media and new media as sources of information has become less meaningful, as the consumers of media have begun to identify these influencers as new information providers.

Furthermore, social media could be used to intervene in suicidal ideation or suicide attempt by encouraging help-seeking behavior that relies on the user's anonymity. Expression of thoughts and intentions about a concerning behavior is stigmatized. The rate of help-seeking behavior for mental issues like suicidality is low due to social stigma. Evidence suggests that "55% of people who complete suicide have no contact with a primary care provider in the month before suicide and 68% have no contact with mental health services in the year before suicide" ([32] as cited in [33]: p.525). SNSs can solve this issue by creating an anonymous online sphere where how people communicate and behave is less influenced by social desirability or social influence.

Many studies suggest that self-disclosure and honesty tend to increase online when participants' identities are hidden. Joinson's study [34] shows that the visual anonymity in computer-mediated communication (CMC) settings heightens the level of self-disclosure. Bargh, Mckenna, and Fitzsimons's experiment [35] also revealed that the likelihood of one's true self being activated is higher in Internet setting than it is in face-to-face setting due to the relative anonymity. Thus, individuals can present themselves in ways that might not be possible in face-to-face settings, thereby promoting help-seeking and collective coping behaviors.

The results of the preliminary coding analysis in this study revealed specific characteristics of content deliverers. There are multiple groups of content uploaders, also known as channel operators, who are distinct from message deliverers. The group of content uploaders are listed as clinic and health organizations, news agencies, one-person creators, production organizations, educational facilities, religious groups, and others, while the group of message deliverers are listed as survivors of suicide attempt; family members of the deceased; friends; news personnel; rescuers; third-party narrators; artists, musicians, and film personnel; lecturers and educators; medical professionals; one-person creators; and others. Some of the message deliverers provide real names, whereas the others are anonymous. Considering the aforementioned characteristics of the content deliverer, the following hypotheses are proposed.

H1. The characteristics of the deliverer, that is, the one who delivers the suicide-themed message would determine the degree of viewer engagement.

H1-1: The degree of engagement with the suicide-themed content would differ according to the content uploader.

H1-2: The degree of engagement with the suicide-themed content would differ according to the message deliverer.

H1-3: The degree of engagement with the suicide-themed content would differ according to the anonymity of the message deliverer.

## Characteristics of the content story

Media contagion effect has been examined from multiple perspectives. A meta-analysis of suicide induced by media identified factors and conditions that maximize or minimize the

copycat effect. These factors include the characteristics of the suicide story (i.e. celebrity or politician vs. non-celebrity, real vs. fictional, and completion vs. attempt), the amount of coverage, period effects (i.e. pre-television era vs. post), characteristics of the suicide rate, and media type (i.e. newspapers vs. television) [11]. Studies that are based on newspapers compared to television (TV, 82% less likely), studies that include suicide stories of political/entertainment celebrity (14.3 times), studies based on real suicides (4.03 times) as opposed to fictional suicides in films and soap operas, and studies based on suicide attempts as an outcome measure as opposed to completed suicide rates or counts are more apt to investigate copycat effects [11]. In addition, media influence on suicide has been studied in multiple country-settings. Regardless of the small effect size compared to other psychosocial risk factors for suicide, media contagion shows that not only audience characteristics but also media content involve risk [36,37].

Since the current study did not consider the difference in viewer engagement among different media or the period effect, only the characteristics of suicide story were included among multiple factors. The results of the preliminary coding analysis in this study categorized content into three story characteristics: (1) celebrity stories, politician stories, and non-celebrity stories; (2) real stories and fictional stories; and (3) suicide attempts, complete suicides, and suicide ideation (see Table 2). Considering the aforementioned characteristics of the suicide-themed content, the following hypotheses are proposed.

H2. The characteristics of the story characteristics, that is, the type of messages delivered, would determine the degree of viewer engagement.

H2-1: The degree of engagement with suicide-themed content would be higher for celebrity suicide stories than for non-celebrity suicide stories.

H2-2: The degree of engagement with suicide-themed content would be higher for real suicide stories than for fictional suicide stories.

H2-3: The degree of engagement with suicide-themed content would be higher for suicide attempt stories than for complete suicide stories and suicide ideation stories.

## Characteristics of content expression

To promote safe media environment, the WHO and national agencies developed guidelines on reporting suicide [38], which includes 11 recommendations as follows: "take the opportunity to educate the public about suicide," "avoid language which sensationalizes or normalizes suicide, or presents it as a solution to problems," "avoid prominent placement and undue repetition of stories about suicide," "avoid explicit description of the method used in a completed or attempted suicide," "avoid providing detailed information about the site of a completed or attempted suicide," "word headlines carefully," "exercise caution in using photographs or video footage," "take particular care in reporting celebrity suicides," "show due consideration for people bereaved by suicide," "provide information about where to seek help," and "recognize that media professionals themselves may be affected by stories about suicide" [38: p. 7]

The rationale for the guidelines is that some reporting characteristics could either prevent or trigger suicides. The guidelines are commonly used as educational material for journalists and editors of traditional media agencies and were developed for traditional news reports of suicide rather than online news or social media posts. Thus, reporters are advised to refrain from using visual material. However, majority of new media content is visual-based, indicating the need to customize existing guidelines based on the new media message or channel features.

Reflecting the reporting guidelines for suicide stories, the current analysis categorized content expression characteristics into five groups: existence of advertisement, expression of

suicide method, existence and placement of warning signs, existence and placement of hot-lines, and the genre category. In this current study, the results of the preliminary coding procedure listed graphic, verbal, and textual expressions of suicide method. Warning signs and hotlines were listed in the description, the first-half of the video clip, and in the second-half of the video clip. YouTube platform provides a special warning function. Only those who have clicked on the "I understand and wish to proceed" option after being shown the YouTube community warning for inappropriate or offensive content warranting viewer discretion are allowed to view the content. The genre categories were listed as Entertainment, People & Blogs, News & Politics, Music, Film & Animation, Nonprofits & Activism, and Education (see Table 3). Considering the aforementioned characteristics of expressive methods, the following hypotheses are proposed.

H3. The characteristics of content expression, that is, how the message is delivered, would determine the degree of viewer engagement.

H3-1: The degree of engagement with suicide-themed content would be higher for advertised videos than for non-advertised videos.

H3-2: The degree of engagement with suicide-themed content would be higher for graphic illustration of suicide method than for verbal or textual illustration of suicide method.

H3-3: The degree of engagement with suicide-themed content would differ according to the existence of a warning sign.

H3-4: The degree of engagement with suicide-themed content would differ according to the position of the warning sign.

H3-5: The degree of engagement with suicide-themed content would differ according to the existence of the hotline.

H3-6: The degree of engagement with suicide-themed content would differ according to the position of the hotline.

H3-7: The degree of engagement with suicide-themed content would differ according to the genre of the content.

## 3. Methods

The current study employed a quantitative content analysis method to identify the factors that draw viewers to suicide-themed videos on YouTube. Content analysis is a research method that examines the characteristics of the content, and it involves a systematic, objective, quantitative analysis of the message characteristics [39]. A thorough exploration of the content, including what the users are exposed to or what kind of messages they are currently acquiring online, from whom, and in what ways the message is received, is the most suitable method for investigation.

The preliminary analysis employed a bottom-up grounded theory approach [40], which is a qualitative method, to examine the characteristics of suicide-themed videos. The sample included a selection of 100 videos from the top to bottom in the order of exposure in the keyword search results with the term "suicide" in English on YouTube. The keyword search was done in Seoul, South Korea at one point in time, September 2019, by the researchers using Incognito Window on Google Chrome browser. The search result was ranked by the default method that YouTube provides which is 'relevance.' Neither specific inclusion nor exclusion criteria was set in the sample selection process with the purpose of extracting all possible codes

relevant to suicide-themed videos. The sample video content was coded to observe the specific instances of the content delivers, the messages, and the expressions. The characteristics of content deliverer were identified by three factors: uploader category, message deliverer category, and anonymity. The characteristics of the stories included four factors: whether the story involves a public or non-public figure, whether the story is real or fictional, whether the story is about a suicide attempt, completed suicide, or suicide ideation, and the number of suicide stories. The characteristics of story expressions were observed using 7 factors: the existence of advertisement, illustration of method, existence of a warning sign, placement of the warning sign, existence of hotlines, placement of hotlines, and genre category. Every newly observed item for each factor was recorded. The list of the items for each factor was built upon after several iterative processes until saturation had been reached. The iterative process continued until only redundant instances were observed and until no new codes occurred to the degree in which the researchers have agreed that further data collection or data coding is counter-productive [41].

Categories were extracted based on the list of characteristics and were used as a foundation for the coding protocols for the quantitative content analysis. In the search results for the keyword "suicide," an additional 100 YouTube videos were retrieved and analyzed. Unlike the search results from search engines, YouTube has no clear distinction in terms of pages. After several videos, a swipe up motion leads to the loading of more videos. Considering that the number of videos presented before the first swipe up motion was 20, it can be regarded that a page on YouTube contains 20 videos. A total of 589 videos were available for the search term 'suicide' after pages loaded until 'No more results' were left to show. Among 589 videos, 100 videos were chosen in the order of exposure after excluding search results for superhero movie based on DC Comics 'Suicide Squad.' Those videos were discernible through the video title and thumbnail, as the major actors and characters were visible in the thumbnail area. The researchers have eliminated 7 Suicide Squad videos because of the low relevance to the health-related issue of suicide, and added 7 other videos to make 100. Videos categorized as 'music' or 'film' were not related to 'Suicide Squad' but they were videos created by individuals who express suicide-related information through the form of music or film.

The sample size (n = 100) was chosen because the purpose of the research was to reflect a basic query of what general people would likely to be exposed to with the keyword search. The first several pages of search result presented to the person searching the keyword engage most viewers whereas the following pages are less attended [42,43]. Multiple health information-related studies included the first several pages of search results in the sample, implying that people tend to select the information they are provided with first rather than the information they are provided later [44–49]. The first two to three pages were the most commonly observed number for YouTube content analysis on health-related matters.

However, a power analysis was performed as Niederkrotenthaler, Schacherl, and Till [50] did to identify the minimum number of samples required. The desired sample size was computed with the software G*Power 3.1 [51]. In order to identify a medium-sized difference effect size ($f^2$) = 0.15; with an Alpha-level of 0.05 and Power (1-β error prob) = 0.80, with the number of predictors n = 3 (who, what, how models), and the total number of predictors 44 (1 continuous variable and 43 dummy variables), a total of 82 videos were required as a minimum. Since each page holds 20 videos, this study required more than four pages to meet the minimum number of samples. Thus, five pages were included in the final sample, in other words, 100 videos.

A comprehensive content analysis was conducted which investigated the following: (1) who uploaded and delivered the suicide-themed videos, (2) what kind of messages were delivered and (3) how the messages were delivered. The data extracted for each video were as follows: (1)

video identification information, which included the title, description, and upload date; (2) characteristics of content deliverer, which included the uploader category, message deliverer category, and anonymity; (3) characteristics of the stories, including whether the story involves a public or non-public figure, whether the story is real or fictional, whether the story is about a suicide attempt, completed suicide, or suicide ideation, and the number of suicide stories; and (4) characteristics of story expressions, which included the existence of advertisement, illustration of method, existence of a warning sign, placement of the warning sign, existence of hotlines, placement of hotlines, and genre category. A total of 14 variables were examined. The characteristics of content deliverer, stories, and expressions were dummy coded.

Furthermore, YouTube's user engagement metrics, including the view count, the number of likes, and the number of comments of the selected 100 videos, were retrieved using YouTube Statistics, which is a free application that tracks the statistics for YouTube videos [52]. It is assumed that the degree of viewer engagement increases in the following order: view count, number of likes, and number of comments. Pressing the like button requires more engagement than merely watching the video, whereas active expression of an opinion through a comment requires additional time and effort. With dummy coding, statistical analysis using hierarchical multiple regression was conducted to observe the linear relationships between the categorical factors of suicide-themed video contents and the amount of attention or popularity.

Hierarchical multiple regression is a method that considers the relative effect of more than one explanatory variable on the dependent variable of interest. It enables the researchers to build several models to compare the proportion of explained variance in the dependent variable by sequentially adding models. The newly added models always include the previous models. The analysis can determine which model better explains and predicts the dependent variable in a statistically meaningful way. The current study takes three models, sometimes referred as blocks: who, what, and how variables of the suicide-themed content in explaining viewer engagement. The analysis was completed on IBM SPSS (Statistical Package for the Social Sciences) Statistics software [53].

## 4. Results

This study hypothesized that deliverer characteristics, story characteristics, and content expression characteristics would predict viewers' attention or engagement with suicide-themed videos. Among 14 variables, 5 variables including message deliverer category, whether the story is about a suicide attempt, completed suicide, or suicide ideation, illustration of method, placement of the warning sign, and placement of hotlines were multi-coded. Thus, the number of instances coded in each category may not be equal to the total number of observed instances which is 100. The descriptive analyses of the main factors are presented in Tables 1–3. The results showed that the regression model with three levels (deliverer characteristics, story characteristics, and content expression characteristics) had different explanatory powers according to the degree of engagement, each measured by the number of views, likes, and comments.

The results of the hierarchical multiple regression are illustrated in Table 4. Deliverer characteristics were the only predictor that was significant across all three viewer responses (p = .036, adjusted $R^2$ = .132 for the view count; p = .011, adjusted $R^2$ = .175 for the number of likes; and p = .048, adjusted $R^2$ = .129 for the number of comments). Hypothesis 1 was partially supported, hypothesis 2 was not supported, and hypothesis 3 was partially supported. In particular, the regression analysis showed that survivors of suicide attempt (β = .338, t = 2.818, p = .006 for the number of likes and β = .443, t = 3.303, p = .001 for the number of comments),

**Table 1. Descriptive statistics for content deliverer characteristics (who).**

| | Category | N | View count | | Number of likes | | Number of comments | |
|---|---|---|---|---|---|---|---|---|
| | | | Mean | SD | Mean | SD | Mean | SD |
| Content Uploader | Clinic and Health Organization | 5 | 542 | 559 | 5 | 7 | 1 | 1 |
| | News Agency | 24 | 1,990 | 2,791 | 15 | 17 | 5 | 7 |
| | One Person Creator | 6 | 6,252 | 11,848 | 369 | 834 | 111 | 260 |
| | Production Organization | 24 | 22,464 | 32,103 | 187 | 209 | 17 | 24 |
| | Educational Facilities | 13 | 988 | 1,798 | 25 | 46 | 2 | 5 |
| | Religious Group | 2 | 1,185 | 1,519 | 6 | 6 | 1 | - |
| | Others | 26 | 5,845 | 6,488 | 124,138 | 162 | 16 | 18 |
| Message Deliverer | Survivors | 11 | 4,398 | 8,844 | 239 | 610 | 64 | 191 |
| | Family | 16 | 845 | 1,177 | 20 | 28 | 3 | 6 |
| | Friends | 7 | 2,571 | 3,969 | 46 | 59 | 9 | 11 |
| | News personnel | 16 | 2,342 | 3,266 | 16 | 19 | 6 | 8 |
| | Rescuer | 2 | 3,534 | 1,167 | 83 | 12 | 9 | 1 |
| | Narrator | 10 | 2,241 | 1,720 | 40 | 37 | 15 | 19 |
| | Artist/Musician/Film personnel | 24 | 24,034 | 31,266 | 218 | 206 | 19 | 23 |
| | Lecturer/Educator | 8 | 330 | 346 | 5 | 5 | - | - |
| | Medical personnel | 8 | 386 | 449 | 4 | 4 | 1 | - |
| | One-person creator | 7 | 10,034 | 12,457 | 445 | 766 | 108 | 237 |
| | Others | 15 | 2,634 | 5,057 | 30 | 56 | 8 | 13 |
| Anonymity | Anonymous | 41 | 12,791 | 20,216 | 151 | 196 | 17 | 22 |
| | Real name provided | 59 | 4,574 | 15,939 | 75 | 276 | 18 | 87 |

Unit: One thousand. Numbers below one thousand are marked as "-".

artists/musicians/film personnel (β = .575, t = 2.653, p = .010 for the number of likes and β = .538, t = 2.098, p = .039 for the number of comments), and one-person creators (β = .423, t = 3.218, p = .002 for the number of likes and β = .414, t = 2.813, p = .006 for the number of comments) were significant predictors.

Although the final model was not a significant predictor of view count and number of likes, it was a significant predictor of the number of comments, which indicates the highest level of

**Table 2. Descriptive statistics for story characteristics (what).**

| | | View count | | Number of likes | | Number of comments | |
|---|---|---|---|---|---|---|---|
| Category | N | Mean | SD | Mean | SD | Mean | SD |
| Celebrity | 7 | 3,386 | 6,554 | 81 | 168 | 9 | 9 |
| YouTuber | 2 | 13,780 | 15,617 | 405 | 514 | 46 | 38 |
| Non-celebrity | 71 | 3,561 | 5,904 | 78 | 249 | 18 | 79 |
| Other | 20 | 56,259 | 48,574 | 366 | 300 | 37 | 38 |
| Real | 61 | 2,743 | 5,036 | 69 | 265 | 18 | 85 |
| Fictional | 13 | 16,710 | 24,510 | 185 | 207 | 18 | 23 |
| Other | 26 | 20,297 | 32,368 | 281 | 342 | 43 | 37 |
| Attempt | 26 | 4,319 | 7,432 | 142 | 405 | 32 | 127 |
| Complete | 41 | 3,016 | 4,734 | 47 | 92 | 9 | 12 |
| Ideation | 40 | 2,802 | 5,189 | 57 | 128 | 8 | 17 |

Unit: One thousand. Numbers below one thousand are marked as "-".

**Table 3. Descriptive statistics for content expression characteristics (how).**

| Category | N | View count Mean | SD | Number of likes Mean | SD | Number of comments Mean | SD |
|---|---|---|---|---|---|---|---|
| Advertisement O | 22 | 20,267 | 29,992 | 182 | 184 | 16 | 21 |
| Advertisement X | 78 | 4,509 | 11,199 | 86 | 262 | 18 | 76 |
| Graphic Expression | 20 | 17,360 | 30,586 | 192 | 235 | 22 | 25 |
| Verbal Expression | 29 | 3,355 | 6,475 | 113 | 380 | 30 | 120 |
| Textual Expression | 5 | 6,007 | 8,207 | 68 | 92 | 18 | 19 |
| None | 57 | 6,810 | 14,379 | 72 | 121 | 9 | 148 |
| Warning sign O | 10 | 4,681 | 6,204 | 99 | 108 | 15 | 19 |
| Warning sign in title | 1 | 1,130 | - | 58 | - | 6 | - |
| Warning sign in the description | 2 | 5,307 | 5,906 | 199 | 199 | 29 | 32 |
| Warning sign in the first-half of the video | 12 | 3,981 | 5,854 | 82 | 105 | 13 | 18 |
| YouTube Warning | 5 | 6,311 | 10,506 | 175 | 332 | 28 | 34 |
| Warning Sign X | 85 | 8,508 | 19,554 | 104 | 258 | 17 | 73 |
| Hotline O | 22 | 4,379 | 7,620 | 146 | 445 | 45 | 145 |
| Hotline in description | 13 | 6,539 | 9,187 | 237 | 581 | 68 | 181 |
| Hotline in the first half of the video | 4 | 1,457 | 2,138 | 20 | 25 | 16 | 28 |
| Hotline in second-half of the video | 15 | 3,874 | 8,142 | 179 | 546 | 62 | 183 |
| Hotline X | 78 | 8,988 | 20,116 | 97 | 97 | 11 | 17 |
| Entertainment | 20 | 15,058 | 31,373 | 156 | 227 | 18 | 23 |
| People & Blogs | 10 | 2,411 | 1,744 | 54 | 44 | 8 | 7 |
| News & Politics | 24 | 1,875 | 2,805 | 14 | 17 | 5 | 7 |
| Music | 10 | 24,119 | 25,094 | 225 | 204 | 18 | 26 |
| Science & Technology | 1 | 1,220 | - | 17 | - | 0 | - |
| Film & Animation | 8 | 14,778 | 12,885 | 204 | 150 | 19 | 15 |
| Gaming | 1 | 3,891 | - | 56 | - | 5 | - |
| Nonprofits &Activism | 14 | 3,041 | 8,021 | 167 | 549 | 47 | 171 |
| Education | 12 | 1,424 | 1,877 | 28 | 38 | 10 | 18 |

Unit: One thousand. Numbers below one thousand are marked as "-".

viewer engagement (p < .001, adjusted $R^2$ = .554). The analysis of the final model showed that educational facilities (β = - 1.987, t = -5.995, p < .001) and religious groups (β = -.949, t = -4.612, p < .001) as the video uploader and rescuer (β = .389, t = 2.911, p = .006) as the deliverer were significant predictors of the number of comments, whereas other deliverer-related variables were not. Hypotheses 1–1 and 1–2 were supported. However, anonymity was not a significant determinant of viewer response, thus rejecting hypothesis 1–3.

None of the story-related variables was significant, rejecting hypotheses 2–1, 2–2, and 2–3. On the other hand, three content expression-related variables were significant predictors of comments: textual expression of suicide method (β = -.230, t = -2.034, p = .048), People and Blogs as genre (β = .425, t = 2.183, p = .034), and Nonprofits & Activism as genre (β = 2.510, t = 7.645, p < .001), supporting hypotheses 3–2 and 3–7. The existence and placement of advertisements, warning signs, and hotlines did not have a significant influence on viewer response, rejecting hypotheses 3–1, 3–3, 3–4, 3–5, and 3–6. Thus, who delivers the suicide-themed-message and how the message is delivered are more significant predictors than what is discussed in terms of viewer engagement.

**Table 4. Results for the hierarchical multiple regression analysis for popularity.**

| Factors (Characteristics) | | Engagement (Standardized Coefficients *beta*) | | | | | | | | |
|---|---|---|---|---|---|---|---|---|---|---|
| | | View count | | | Number of Likes | | | Number of Comments | | |
| | | Model 1 | Model 2 | Model 3 | Model 1 | Model 2 | Model 3 | Model 1 | Model 2 | Model 3 |
| Content Deliverer¶ (Who) | Clinic and health organization | -.104 | -.111 | -0.186 | 0.029 | 0.002 | -0.096 | 0.073 | 0.062 | -0.048 |
| | News agency | -.218 | -.128 | 0.066 | 0.068 | 0.126 | 0.401 | 0.122 | 0.192 | 0.465 |
| | One-person creator | -.152 | -.023 | 0.024 | 0.155 | 0.299 | 0.128 | 0.298 | 0.454 | 0.092 |
| | Educational facilities | -.186 | -.041 | -0.238 | 0.034 | 0.121 | -0.778 | 0.073 | 0.159 | -1.987*** |
| | Religious groups | -.060 | -.011 | -0.154 | 0.021 | 0.067 | -0.321 | 0.015 | 0.074 | -0.949*** |
| | Others | -.166 | -.095 | -0.135 | 0.125 | 0.195 | 0.148 | 0.134 | 0.266 | -0.049 |
| | Survivors | -.046 | -.045 | 0.1 | 0.338** | 0.293 | 0.311 | 0.443*** | 0.418 | -0.042 |
| | Family | -.113 | -.102 | -0.102 | 0.049 | 0.032 | -0.056 | 0.127 | 0.111 | -0.173 |
| | Friends | .012 | .021 | 0.069 | 0.021 | 0.042 | 0.055 | -0.006 | -0.009 | 0.034 |
| | News personnel | -.025 | -.014 | 0.004 | 0.119 | 0.123 | 0.045 | 0.237 | 0.286 | -0.177 |
| | Rescuer | -.027 | -.019 | 0.077 | 0.09 | 0.059 | 0.31 | 0.122 | 0.087 | 0.389** |
| | Narrator | .006 | -.021 | -0.035 | 0.125 | 0.061 | -0.21 | 0.291 | 0.359 | -0.05 |
| | Artist/Musician/Film personnel | .376 | .286 | 0.183 | 0.575** | 0.541 | 0.271 | 0.538* | 0.742 | 0.121 |
| | Lecturer/Educator | -.073 | -.098 | -0.017 | 0.1 | 0.053 | 0.08 | 0.178 | 0.171 | -0.235 |
| | Medical personnel | -.09 | -.052 | -0.006 | 0.012 | 0.078 | 0.102 | 0.035 | 0.167 | 0.028 |
| | One-person creator | .107 | .024 | -0.147 | 0.423** | 0.31 | 0.193 | 0.414** | 0.395 | 0.136 |
| | Others | -.011 | .042 | 0.039 | 0.073 | 0.09 | 0.196 | 0.14 | 0.153 | 0.036 |
| | Anonymity | -.190 | -.171 | -0.205 | -0.009 | -0.062 | -0.075 | -0.023 | -0.023 | 0.038 |
| Content Story¶ (What) | Number of stories | | -.049 | -0.077 | | -0.072 | -0.007 | | -0.031 | 0.06 |
| | Celebrity | | -.014 | -0.03 | | -0.099 | -0.076 | | -0.236 | -0.003 |
| | YouTuber | | .114 | 0.027 | | 0.175 | 0.164 | | 0.028 | 0.031 |
| | Other | | .215 | 0.244 | | 0.031 | 0.05 | | 0.045 | 0.144 |
| | Fictional | | -.029 | -.059 | | 0.033 | 0.137 | | -0.132 | -0.029 |
| | Attempt | | -.032 | 0.095 | | 0.069 | -0.129 | | 0.11 | 0.166 |
| | Complete | | -.099 | 0.092 | | -0.043 | -0.026 | | 0.039 | 0.136 |
| | Ideation | | -.103 | -0.128 | | -0.078 | -0.091 | | -0.05 | 0.108 |
| Content Express-ion¶ (How) | Advertise-ment | | | -0.003 | | | 0.024 | | | -0.052 |
| | Graphic expression of method | | | 0.078 | | | 0.025 | | | 0.144 |
| | Verbal expression of method | | | -0.038 | | | -0.116 | | | 0.261 |
| | Textual expression of method | | | -0.229 | | | -0.122 | | | -0.23* |
| | Warning sign Existence | | | 0.373 | | | 0.1 | | | 0.2 |
| | in Title | | | 0.097 | | | -0.03 | | | -0.167 |
| | in description | | | 0.192 | | | 0.117 | | | 0.232 |
| | in the first-half of the video | | | 0.189 | | | -0.044 | | | -0.025 |
| | Hotline Existence | | | | | | -0.048 | | | 0.168 |
| | in the description | | | 0.078 | | | 0.23 | | | 0.018 |
| | in the first-half of the video | | | -0.038 | | | -0.114 | | | -0.017 |
| | in the second-half of the video | | | -0.229 | | | 0.061 | | | 0.05 |
| | Entertainment | | | 0.373 | | | 0.339 | | | 0.251 |
| | People & Blogs | | | 0.097 | | | 0.183 | | | 0.425* |
| | Music | | | 0.192 | | | 0.297 | | | 0.167 |
| | Film & Animation | | | 0.189 | | | 0.223 | | | 0.065 |
| | Gaming | | | 0.069 | | | 0.001 | | | -0.017 |
| | Nonprofits &Activism | | | 0.386 | | | 1.298 | | | 2.51** |
| | Education | | | 0.234 | | | 0.43 | | | 0.168 |

(*Continued*)

**Table 4.** (Continued)

| Factors (Characteristics) | Engagement (Standardized Coefficients *beta*) | | | | | | | | |
| --- | --- | --- | --- | --- | --- | --- | --- | --- | --- |
| | View count | | | Number of Likes | | | Number of Comments | | |
| | Model 1 | Model 2 | Model 3 | Model 1 | Model 2 | Model 3 | Model 1 | Model 2 | Model 3 |
| R Square | .292 | .341 | .455 | .327 | .364 | .595 | .300 | .344 | .782 |
| Adjusted R Square | .132 | .090 | -.048 | .175 | .122 | .222 | .129 | .072 | .554 |
| F | 1.827* | 1.358 | .905 | 2.155* | 1.503 | 1.595 | 1.759* | 1.264 | 3.427** |
| Durbin-Watson | | | 2.016 | | | 2.006 | | | 1.849 |

* $p < .05$

** $p < .01$

*** $p < .001$.

## 5. Discussion and conclusion

The findings showed that videos uploaded by educational facilities and religious groups, and videos textually expressing suicide methods had relatively fewer comments. On the contrary, videos categorized as People & Blogs and Non-profits & Activism, and content delivered by rescuers of suicide had relatively more comments. Content delivered by survivors of suicide attempt, artists, musicians, film personnel, and one-person creators also received more likes and comments.

These findings imply that viewers are more engaged with the content when the deliverers have close experience of suicide. Sharing suicide stories through art, music, and film such as in vlog format is involving, whereas lecture-based or preaching approaches are less involving. Traditionally, suicide has been discussed by suicide prevention organizations or medical professionals via news channels. Suicide stories have been publicized through news portals, and the influence of publicized suicide stories has been studied. Although it is difficult to deny the influence of informative and educational news content, this study shows that suicide conversations are carried out in online spheres, where rescuers and survivors of suicide attempt actively participate in the discussion. Suicide prevention organizations and educational facilities need to strategically engage viewers.

This study has implications for not only health and medical professionals but also platform service providers. As the deliverers of suicide-themed posts are survivors and rescuers rather than health professionals, the platform may play an important role as an arena for diagnosis. The symptoms and the reasons for suicide ideation may be explicitly stated on the platform, which may help health professionals to diagnose individuals who ideate suicide or those with suicide experiences. The extant studies suggest that individuals can positively cope with suicidal thoughts when they openly talk about their state of mind. Much of the video content analyzed in this study addressed the need for a platform to discuss suicide and to share personal feelings without judgement and stigma. As these dominant media platforms are becoming the locus for open discussion, education, and collective coping, platform service providers are recommended to continue facilitating the discussion by engaging more people.

The accountability of participants in suicide-themed online conversations should be equally emphasized as much as the accountability of platforms. The creators of suicide-themed videos and the viewers should take advantage of the platform, but with discretion. Uploaders should acknowledge the societal influence of their posts, and the viewers should actively alert the authorities of any harmful or triggering content.

Most importantly, this study also has implications for policymakers in terms of addressing the need for developing a proper guideline for suicide-themed new media content. The current

guidelines include news reporting guidelines, which advise reporters to refrain from using visual material. However, a majority of new media content includes visual expressions. The current analysis showed that over 50% of suicide-themed content on YouTube involves graphic, verbal, or textual illustrations of suicide methods, while the majority did not provide any warning sign or crisis hotlines. Only 5% of the observed content required age registration by the platform, which has been highlighted as a problem [50]. As young adults tend to obtain information and resources through online channels, new media platforms might be the first or most-frequently visited sources of information. Therefore, there is a need to revise the existing guidelines to fit new media features.

This study is limited in that only content-related predictors were included in the analysis. Platform affordances or external factors were not taken into consideration. Moreover, alternative measurement of viewer engagement should be considered, such as the positive comment to negative comment ratio or the net comment calculated by the number of positive comments subtracted by the number of negative comments. A more appropriate measure of viewer engagement other than the number of views, likes, and comments will provide more fruitful implications as positive engagement on sensitive topics like suicide enables collective coping. In addition, the video samples analyzed in this study were search results, which had already been filtered by the platform. This indicates that extremely triggering or harmful content had already been removed from the website, which were, therefore, not included in the analysis. However, it can be concluded that the sample did consist of videos maintained available on the platform that an ordinary user would find using the same keyword. Lastly, the applicability of research results can be another limitation since selecting a certain number of videos at a specific time in a specific location with specific language may not incorporate all instances of suicide-themed videos on YouTube. Nonetheless, the selection of top several pages in the order of relevance was the best alternative as the formula of search result presentation on YouTube is unknown like a black box. Follow up studies pertaining to multiple location and language settings can be helpful.

In order to determine whether dominant visual media platforms facilitate the diffusion of suicidal thoughts, future studies are needed to identify the factors of dominant visual media platform that augment or spread suicidal thoughts. As society continues to undergo digital transformation, daily-visited new media sites, such as Twitter, Facebook, and YouTube, should not facilitate suicide, but help to mitigate suicidal thoughts. The findings of this germinal explorative study could help establish a cornerstone for a safe online community and a constructive communication ground, by examining societal issues and highlighting the responsibilities of dominant visual new media platforms.

## Supporting information

**S1 Codebook.**
(DOCX)

**S1 File.**
(XLSX)

**S2 File.**
(DOCX)

## Author Contributions

**Supervision:** Seongcheol Kim.

**Writing – original draft:** Eun Ji Jung.

**Writing – review & editing:** Seongcheol Kim.

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
