## [Decision Letter · Decision Letter 0]

23 Mar 2021

PONE-D-20-36601

Suicide on YouTube: Factors engaging viewers to suicide-themed videos

PLOS ONE

Dear Dr. Kim,

Thank you for submitting your manuscript to PLOS ONE. After careful consideration, we feel that it has merit but does not fully meet PLOS ONE’s publication criteria as it currently stands. Therefore, we invite you to submit a revised version of the manuscript that addresses the points raised during the review process.

We look forward to receiving your revised manuscript.

Kind regards,

Vincenzo De Luca

Academic Editor

PLOS ONE

Journal Requirements:

Reviewers' comments:

Reviewer's Responses to Questions

**Comments to the Author**

1. Is the manuscript technically sound, and do the data support the conclusions?

Reviewer #1: Partly

Reviewer #2: No

2. Has the statistical analysis been performed appropriately and rigorously? 

Reviewer #1: Yes

Reviewer #2: Yes

3. Have the authors made all data underlying the findings in their manuscript fully available?

Reviewer #1: Yes

Reviewer #2: Yes

4. Is the manuscript presented in an intelligible fashion and written in standard English?

Reviewer #1: Yes

Reviewer #2: Yes

5. Review Comments to the Author

Reviewer #1: 1. Summary of the research and your overall impression

Dear Authors,

Thank you for the privilege of reading your interesting work, Suicide on YouTube: Factors engaging viewers to suicide-themed videos.

In this work, you examine what factors are correlated with viewer engagement for a sample of 100 YouTube videos resulting from the keyword search “suicide”. You examine three groups of factors. “Characteristics of the deliver” (e.g. health professional vs survivor), “characteristics of the content story” (real vs fictional), and “characteristics of content expression” (e.g. genre, warning signs). Engagement factors include views, likes and comments. You then conduct hierarchical multiple regression to quantify the relation between these factors and engagement. You present your results, where you find some aspects of content delivery and expression are significantly correlated with greater or lesser engagement. Your conclusion discusses the results, and how they might apply to various groups such as platforms and health professionals.

I find general strengths of this work include addressing a gap in the literature in an important and applicable subject area; there does not seem to be a lot of work looking into suicidal content of youtube videos, despite their possible impact on an important health outcome. The paper also does a good job of reviewing some older literature, and has some methodological strengths such as you come up with the categories and their factors. Weaknesses include much of the methodology being poorly described, which makes it hard to assess the validity of some of the claims in the paper at this time.

I believe major revisions are warranted in order to address some of the above weaknesses. Please see my comments below for further details.

2. Discussion of specific areas for improvement

Major Issues:

1. The title of the paper, and various points within it, suggest that the study can draw conclusions about suicide videos on YouTube in general. However, only 100 videos are sampled, at one point in time. My understanding is that youtube video search results can very greatly based on the youtube account searched from, time, geography, and other factors. So, I am not sure how representative these 100 videos are, and whether general conclusions can be drawn.

a. Description of this search is required; e.g. where it was searched, when it was searched, and by whom. Additionally, you need to include how it was ranked; the default method I believe is “relevance” but it can also be by view count, like ratio, etc.

b. I believe having the study as an examination of 100 videos at a given point of time as searched by one person is probably still interesting and interpretable, but I believe your paper’s word choice should reflect this. E.g. title could be “…Factors engaging viewers on a sample of suicide-themed videos”, and generally the paper should acknowledge you are only examining one sample of 100 videos.

c. Some discussion about how persistent search results are would be helpful. If someone else searching for “suicide” a week later would get an entirely different set of 100 videos, are the results of your paper still useful? I tested briefly searching with two different Youtube/gmail accounts logged on, and found that there were some different videos resulting each time, though the majority were the same. A general audience unfamiliar with Youtube will want to know how applicable your results are.

d. You do not explain why 100 videos was the number chosen. It would be helpful to know how many videos are out, so we could know how representative this sample is. For example, you find religious organization videos are less engaged with – but what if religious videos in the 100-200th spots are the most engaged? Consider including how many videos might be out there in total, or how many are watched with a certain amount of views e.g. at least 1000 views. Please also see if there is literature discussing what rank of videos usually engage in; if the top 100 videos are usually what 99% of engagement is in, then your sample would be a lot more representative than if it’s only, say, 1%.

e. In summary, the sentence in the paper “However, it can be concluded that the sample consisted of videos that an ordinary user would find using the same keyword” needs to be further substantiated.

2. An important reference for your paper is the WHO 2008 “Preventing Suicide A Resource for Media Professionals” as you use this to determine the content expression characteristic . I believe you are missing this reference in your bibliography, so I assume it is this document you are referring to. However, this was updated in 2017 “Preventing suicide: a resource for media professionals - update 2017” by the WHO.

a. Please include a citation for the report you are using.

b. I believe you should be using the updated 2017 report for your study. On initial glance, your categories may still be applicable given the new update. However, given the importance of this reference, please consider incorporating any necessary changes into your paper.

3. I believe a general strength of the paper is being inclusive of all videos that result from a search. However, when I tried out such a search myself, the top 100 did seem to include at least five videos that were likely not related very much to suicide, such as trailers for the 2016 superhero movie “suicide squad”, and one about a type of car door called “suicide doors” named such because they led to accidental (not intentional) deaths in the past. This makes me wonder if your results are being affected by videos that have very little to do with suicide.

a. The paper does describe how many videos are “music” or “film”.

b. However, I think some discussion of the videos being included would be helpful, especially given that the sample size is not that big. If a basic filtering to remove results clearly not related to suicide is not performed e.g. “suicide doors”, then this should be acknowledged/discussed and perhaps quantified. If no filtering at all was done, please further substantiate and explain the impact of this choice.

Minor issues:

1. I don’t believe data availability is discussed in the paper; the form says it will be in the supplement but this was not available in my manuscript. It may be beneficial to add some details about the data you’ll provide to aid replication?

2. In the major issues section, I discuss how adding further details regarding methodology would be important. Additional areas of methodology should also be described more. Your statistical analysis is not something known by a general audience, and should be explained at least in summary. Additionally, you did not mention how the analysis was performed, including what software was used and any parameters. This is helpful for replication and extension. Discussing why you chose this method, vs other methods, may also be interesting and helpful to add.

3. The authors seem to generally do a good job of citing relevant prior work, and mention the lack of studies look at suicide-related videos on youtube. However, I was able to a few studies that do look quite related published recently in 2020, e.g. High viewership of videos about teenage suicide on YouTube by Dagar and Falcone, and Communication about suicide in YouTube videos: Content analysis of German-language videos retrieved with method-and help-related search terms by Niederkrotenthaler et al. It may be helpful to review and mention these works. This reviewer has no connection to these works or their authors.

4. Table 1 should likely contain median values, especially for the smaller groups where they may be some variation. Alternatively, the authors could consider incorporating graphics such as boxplots to describe the data. I find it a bit hard to read due to the large numbers. If continuing to use numbers, describing the numbers as the nearest thousand (e.g. 1990059 to 1990) might make the numbers easier to compare.

5. Some of the categories add up to more than 100, so I believe some categories can have multiple values. Please address in methodology if this is correct, or what happens if a category is unclear, or multi-valued e.g. a health professional who is also a survivor.

6. Thank you for addressing that you did not examine the like vs dislike ratio in your paper and it would be appropriate for further work. If you have the data readily available, I believe this could be a helpful addition to this paper as another engagement metric that may be quite different than others.

7. I would recommend a different word choice for the sentence “This study focused on one of the self-induced health concerns worldwide” in the abstract. In this context I believe it could be beneficial to describe it more directly as a result of mental health concerns, to emphasise that it is usually due to external factors rather than an individual “choice” e.g. the APA describes it as “Suicide is the act of killing yourself, most often as a result of depression or other mental illness”. Consider other choices such as “focused on a leading cause of death” or “a leading cause of death related to mental health”.

8. Please reconsider or further elaborate on the sentences “As the deliverers of suicide-themed posts are survivors and rescuers rather than health professionals, the platform may play an important role as an arena for diagnosis. The symptoms and the reasons for suicide ideation may be explicitly stated on the platform, which may help health professionals to diagnose individuals who ideate suicide or those with suicide experience”. Has any prior work investigated this? Is there a clinical group (teens?) that posts videos about suicide often enough that this could be clinically useful? Doesn’t youtube already have a “report” button that allows something like this to happen, without the health professionals needing to view the videos directly? As a health professional, this strikes me as too far a jump without a bit more substantiation.

Thank you again for being able to read your work, and I hope you find my feedback is helpful.

Reviewer #2: The paper targets a very interesting topic within social media. Nevertheless it requires major modifications

- The authors should provide sufficient information on the following:

1. The language and region information of the videos analysed. Do any of the videos require age registration?

2. Was the term "Suicide" searched in English? When was the search and video selection performed?

3. Were the browser cache and history cleared before each search and all filters switched off?

- The authors stated that no exclusion criteria were set. It will be useful to exclude unrelated contents (e.g. Music Videos, Playlists, etc.) and/or videos with a length of >10 minutes.

- The "Introduction" and "Literature review and research questions" sections are lengthy and contain redundant information.

6. PLOS authors have the option to publish the peer review history of their article (what does this mean?). If published, this will include your full peer review and any attached files.

Reviewer #1: **Yes: **John-Jose Nunez

Reviewer #2: No

---

## [Author Response · Author response to Decision Letter 0]

7 May 2021

Reviewer #1: 

1. Summary of the research and your overall impression

Dear Authors,

Thank you for the privilege of reading your interesting work, Suicide on YouTube: Factors engaging viewers to suicide-themed videos.

In this work, you examine what factors are correlated with viewer engagement for a sample of 100 YouTube videos resulting from the keyword search “suicide”. You examine three groups of factors. “Characteristics of the deliverer” (e.g. health professional vs survivor), “characteristics of the content story” (real vs fictional), and “characteristics of content expression” (e.g. genre, warning signs). Engagement factors include views, likes and comments. You then conduct hierarchical multiple regression to quantify the relation between these factors and engagement. You present your results, where you find some aspects of content delivery and expression are significantly correlated with greater or lesser engagement. Your conclusion discusses the results, and how they might apply to various groups such as platforms and health professionals.

I find general strengths of this work include addressing a gap in the literature in an important and applicable subject area; there does not seem to be a lot of work looking into suicidal content of youtube videos, despite their possible impact on an important health outcome. The paper also does a good job of reviewing some older literature, and has some methodological strengths such as you come up with the categories and their factors. Weaknesses include much of the methodology being poorly described, which makes it hard to assess the validity of some of the claims in the paper at this time.

I believe major revisions are warranted in order to address some of the above weaknesses. Please see my comments below for further details.

(Response)

Thank you for giving us an opportunity to revise our paper. We are deeply grateful for your insightful and constructive comments. We have taken advantage of these comments in carefully preparing this revision. Added or revised parts were highlighted in the revised manuscript with track changes. Please see our detailed explanations (in blue color) in the individual responses to your comments (in black color). 

2. Discussion of specific areas for improvement

Major Issues:

1. The title of the paper, and various points within it, suggest that the study can draw conclusions about suicide videos on YouTube in general. However, only 100 videos are sampled, at one point in time. My understanding is that youtube video search results can very greatly based on the youtube account searched from, time, geography, and other factors. So, I am not sure how representative these 100 videos are, and whether general conclusions can be drawn.

a. Description of this search is required; e.g. where it was searched, when it was searched, and by whom. Additionally, you need to include how it was ranked; the default method I believe is “relevance” but it can also be by view count, like ratio, etc.

(Response)

Thank you for your valuable comment. Search was done in Seoul, South Korea at one point in time, September 2019, by the researchers, using Chrome browser. The language was set as ‘English (US)’ and the location was set as ‘United States.’ The search result was ranked by the default method “relevance.” Please see our revision (in highlighted parts) in page 16.

b. I believe having the study as an examination of 100 videos at a given point of time as searched by one person is probably still interesting and interpretable, but I believe your paper’s word choice should reflect this. E.g. title could be “…Factors engaging viewers on a sample of suicide-themed videos”, and generally the paper should acknowledge you are only examining one sample of 100 videos.

(Response) 

Thank you for your valuable comment. To respond to your comment, we have changed the title into “Suicide on YouTube: Factors engaging viewers to a selection of suicide-themed videos” as suggested. Please see the new title in page 1 of our revision.

c. Some discussion about how persistent search results are would be helpful. If someone else searching for “suicide” a week later would get an entirely different set of 100 videos, are the results of your paper still useful? I tested briefly searching with two different Youtube/gmail accounts logged on, and found that there were some different videos resulting each time, though the majority were the same. A general audience unfamiliar with Youtube will want to know how applicable your results are.

(Response) 

Thank you for your insightful comment. It is true that YouTube search results can be different according to the owner of the account, geography, and so on. Thus, we opened the Google Chrome browser as Incognito Window, which enables private browsing without having to log in. Nevertheless, considering your comment, we have added the applicability of the sample as one of the limitations of the study. Please see our revision (in highlighted parts) in page 29.

Selecting a certain number of videos at a specific time in a specific location with specific language may not incorporate all instances of suicide-themed videos on YouTube. However, reviewing the first two to three pages can be representative enough because they are “most accessible and easiest to find, and therefore the most representative of what the average consumer would view” (Bae and Baxter, 2018: p. 1942).

Bae SS, Baxter S. YouTube videos in the English language as a patient education resource for cataract surgery. International ophthalmology. 2018; 38(5): 1941-1945. doi: 10.1007/s10792-017-0681-5

In the end, YouTube’s search result suggestion is like a black box. Since the formula of search result presentation on YouTube is unknown, the selection of top several pages in the order of relevance was the best alternative, as many other studies have done the same. 

d. You do not explain why 100 videos was the number chosen. It would be helpful to know how many videos are out, so we could know how representative this sample is. For example, you find religious organization videos are less engaged with – but what if religious videos in the 100-200th spots are the most engaged? Consider including how many videos might be out there in total, or how many are watched with a certain amount of views e.g. at least 1000 views. Please also see if there is literature discussing what rank of videos usually engage in; if the top 100 videos are usually what 99% of engagement is in, then your sample would be a lot more representative than if it’s only, say, 1%.

(Response) 

Thank you for pointing out the importance of explaining the representativeness of the sample. We agree that we should provide more information on why only 100 videos were the number chosen for the final analysis. There were two different reasons behind the selection of the number of videos chosen for each of the analysis.

For the preliminary analysis, the number (n=100) was chosen because it was the point where saturation has been reached. The researchers have agreed that the iterative process can stop. No new codes that could be meaningful to the establishment of coding protocols were generated. The researchers have agreed that the continuation of analysis may lead to the waste of resources, thus, the iterative process has come to a cease. Please refer to line 365 to 370 in page 17. 

“Every newly observed item for each factor was recorded. The list of the items for each factor was built upon after several iterative processes until saturation had been reached. The iterative process continued until only redundant instances were observed and until no new codes occurred to the degree in which the researchers have agreed that further data collection or data coding is counter-productive (Saunders et al., 2018).”

For the final analysis, 100 videos were chosen because the purpose of the research was to reflect a basic query of what general people would likely to be exposed to with the keyword “suicide.” This research suggests that the first several pages of search results presented to the person searching the keyword engage most viewers. 

Previous surveys developed by iProspect, a performance-driven digital marketing agency, and Jupiter Research indicate that people tend to click what appears within the first page of results when using search engines. In the whitepaper, it says “62% of search engine users click on a search result within the first page of results, and a full 90% of search engine users click on a result within the first three pages of search results” (iProspect, 2006). A qualitative study by Eysenbach and Köhler (2002) also suggests that search results on the second or following pages were less attended by the searchers searching for health-related information. This implies that people tend to select the information they are provided with first rather than the information they are provided later. The former work (iProspect Survey) has been cited in multiple health information-related studies that included the first several pages in the sample:

Kelly-Hedrick, Grunberg, Rochu, and Zelkowitz’s 2018 research with keyword search “infertility” where 80 top-viewed YouTube videos were included for the analysis, including the first 4 pages of results (20 results per page)

Stellefson, Chaney B, Ochipa, Chaney D, Haider, Hanik, Chavarria, and Bernhardt’s 2014 work with keyword search “Chronic Obstructive Pulmonary Disease,” “COPD,” “COPD management,” and “COPD self-management” where 223 unique videos were saved for analysis, including the first 7 pages of results

Wasserman, Baxter, Rosen, Burnstein and Halverson’s work in 2014, screened Web site links on the first 2 pages of search (18 to 20 links per page)

Sahin AN, Sahin AS, Schwenter and Sebajang’s 2019 study where the first two pages of search results were reviewed for the keyword search “colorectal cancer,” “colon cancer,” and “bowel cancer.”

Unlike the search results from search engines, YouTube has no clear distinction in terms of pages. After several videos, a swipe up motion leads to the loading of more videos. Considering that the number of videos presented before the first swipe up motion was 20, it can be regarded that a page on YouTube contains 20 videos. A total of 589 videos were available for the search term “suicide” after pages loaded until ‘No more results’ were left to show. 

The first two to three pages were the most commonly observed sample for YouTube content analysis on health-related matters. However, a power analysis was performed as Niederkrotenthaler, Schacherl, and Till did to identify the minimum number of samples required. The desired sample size was computed with the software G*Power 3.1 (Faul, Erdfelder, Buchner, and Lang, 2009). In order to identify a medium-sized difference effect size (f²)=0.15; with an Alpha-level of 0.05 and Power (1-β error prob)=0.80, with the number of predictors n=3 (who, what, how), and the total number of predictors 44 (1 continuous variable and 43 dummy variables), a total of 82 videos were required as a minimum. Since each page holds 20 videos, this study required more than four pages to meet the minimum number of samples. Thus, five pages were included in the final sample, in other words, 100 videos. Niederkrotenthaler, Schacherl and Till’s study in 2020 included less than 100 videos for each search term. The sample size (n=100) should be fine, considering the relevance of the videos on the first three pages. Please see our revision (in highlighted parts) in pages 17~18.

iProspect. iProspect Search Engine User Behavior Study. 2006. Available from: http://district4.extension.ifas.ufl.edu/Tech/TechPubs/WhitePaper_2006_SearchEngineUserBehavior.pdf

Eysenbach G, Köhler C. How do consumers search for and appraise health information on the world wide web? Qualitative study using focus groups, usability tests, and in-depth interviews. BMJ. 2002; 324: 573–577. doi: 10.1136/bmj.324.7337.573

Kelly-Hedrick M, Grunberg PH, Brochu F, Zelkowitz P. “It’s totally okay to be sad, but never lose hope”: content analysis of infertility-related videos on YouTube in relation to viewer preferences. Journal of medical Internet research. 2018; 20(5): e10199. doi: 10.2196/10199

Stellefson M, Chaney B, Ochipa K, Chaney D, Haider Z, Hanik B, et al. YouTube as a source of chronic obstructive pulmonary disease patient education: A social media content analysis. Chronic respiratory disease. 2014; 11(2): 61-71. doi: 10.1177/1479972314525058

Wasserman M, Baxter NN, Rosen B, Burnstein M, Halverson AL. Systematic review of internet patient information on colorectal cancer surgery. Dis Colon Rectum. 2014; 57(1): 64–69. doi: 10.1097/DCR.0000000000000011

Sahin AN, Sahin AS, Schwenter F, Sebajang H. YouTube videos as a source of information on colorectal cancer: what do our patients learn?. Journal of Cancer Education. 2019; 34(6): 1160-1166. doi: 10.1007/s13187-018-1422-9

Leong AY, Sanghera R, Jhajj J, Desai N, Jammu BS, Makowsky MJ. Is YouTube useful as a source of health information for adults with type 2 diabetes? A South Asian perspective. Canadian journal of diabetes. 2018; 42(4): 395-403. doi: 10.1016/j.jcjd.2017.10.056

Bae SS, Baxter S. YouTube videos in the English language as a patient education resource for cataract surgery. International ophthalmology. 2018; 38(5): 1941-1945. doi: 10.1007/s10792-017-0681-5

e. In summary, the sentence in the paper “However, it can be concluded that the sample consisted of videos that an ordinary user would find using the same keyword” needs to be further substantiated.

(Response) 

Thank you for your comment. To respond to your comment, we have changed the sentence to “However, it can be concluded that the sample consisted of videos maintained available on the platform that an ordinary user would find using the same keyword.” Please see our revision in line 549 of page 29. 

Before this sentence, we described that extremely triggering or harmful content had already been removed by the platform because teams like YouTube’s ‘Trust and Safety team’ actively review ‘flagged’ content. This can be the limitation of the sample in that only the remaining videos can be observed, unless timely searched (searched ‘before’ the content has been put down or hidden by the platform). The flagged or removed content may or may not be more triggering than what is left on the platform. Since it is impossible to predict when those potentially ‘flagged’ content will be uploaded and to include those videos in the sample, observing what is remaining on the platform has been the best option. Thus, the observed videos are still meaningful in that they are similar to what an ordinary user would find because those videos are the ones maintained available, still accessible to ordinary users.

2.An important reference for your paper is the WHO 2008 “Preventing Suicide A Resource for Media Professionals” as you use this to determine the content expression characteristic . I believe you are missing this reference in your bibliography, so I assume it is this document you are referring to. However, this was updated in 2017 “Preventing suicide: a resource for media professionals - update 2017” by the WHO.

a. Please include a citation for the report you are using.

(Response) 

As suggested, the WHO 2008 report was added to our reference. In addition, we have reviewed the updated 2017 version and have referred to the report. 

b. I believe you should be using the updated 2017 report for your study. On initial glance, your categories may still be applicable given the new update. However, given the importance of this reference, please consider incorporating any necessary changes into your paper.

(Response) 

Thank you for the update. The content of the two reports is very similar, suggesting guidelines for the media professionals. The main difference was that the latest version acknowledges the helpful impact of responsible reporting better than the former version. ‘Digital media as a double-edged sword in terms of suicide-related information’ has been the very essence of the manuscript. The 2017 updated report has been an additional support for our paper.

3.I believe a general strength of the paper is being inclusive of all videos that result from a search. However, when I tried out such a search myself, the top 100 did seem to include at least five videos that were likely not related very much to suicide, such as trailers for the 2016 superhero movie “suicide squad”, and one about a type of car door called “suicide doors” named such because they led to accidental (not intentional) deaths in the past. This makes me wonder if your results are being affected by videos that have very little to do with suicide.

a. The paper does describe how many videos are “music” or “film”.

b. However, I think some discussion of the videos being included would be helpful, especially given that the sample size is not that big. If a basic filtering to remove results clearly not related to suicide is not performed e.g. “suicide doors”, then this should be acknowledged/discussed and perhaps quantified. If no filtering at all was done, please further substantiate and explain the impact of this choice.

(Response) 

Thank you for your valuable comment. The video selection process did not have exclusion criteria for the preliminary analysis. On the other hand, videos related to the superhero movie “Suicide Squad” were not included in the final analysis. This study explores suicide-themed videos, which are different from videos with the word ‘suicide.’ In order to examine suicide relevant videos, “Suicide Squad” videos were excluded. The authors have added information regarding the exclusion criteria for the final sample. “Suicide Squad” videos were discernible through the video title and thumbnail, as the major actors and characters were visible in the thumbnail area. The researchers have eliminated 7 “Suicide Squad” videos, and had to add 7 other videos to make 100. Videos categorized as ‘music’ or ‘film’ were not related to “Suicide Squad” but they were videos created by individuals who express suicide-related information through the form of music or film. Other videos like ‘suicide doors’ were included because those videos were difficult to exclude unless the content of the video was watched. Please see our revision in line 379 ~ 386 of pages 17~ 18. 

Minor issues:

1.I don’t believe data availability is discussed in the paper; the form says it will be in the supplement but this was not available in my manuscript. It may be beneficial to add some details about the data you’ll provide to aid replication?

(Response) 

Thank you for this comment. We have provided all relevant data underlying the findings within the manuscript. To aid replication, we would like to provide our codebook, so that any researcher who is interested in the further study may use it with his/her own sample.

2. In the major issues section, I discuss how adding further details regarding methodology would be important. Additional areas of methodology should also be described more. Your statistical analysis is not something known by a general audience, and should be explained at least in summary. Additionally, you did not mention how the analysis was performed, including what software was used and any parameters. This is helpful for replication and extension. Discussing why you chose this method, vs other methods, may also be interesting and helpful to add.

(Response) 

Thank you for this valuable comment. The purpose of the study was to figure out which factor engages viewers to suicide-themed videos. In order to find the answer to the research question, it was necessary to incorporate multiple variables into consideration. Since hierarchical multiple regression enables the researchers to build several models to compare the proportion of explained variance in the dependent variable by sequentially adding models, this method was chosen.

We have elaborated the details regarding methodology in the manuscript, providing 1) a brief summary, 2) information on the software used, and 3) a reference material (step-by-step how-to guide for hierarchical linear regression) that can be helpful for the readers. Please refer to line 429 to 437, which reads:

Hierarchical multiple regression is a method that considers the relative effect of more than one explanatory variable on the dependent variable of interest. It enables the researchers to build several models to compare the proportion of explained variance in the dependent variable by sequentially adding models. The newly added models always include the previous models. The analysis can determine which model better explains and predicts the dependent variable in a statistically meaningful way. The current study takes three models, sometimes referred as blocks: who, what, and how variables of the suicide-themed content in explaining viewer engagement. The analysis was completed on IBM SPSS (Statistical Package for the Social Sciences) Statistics software (SAGE Publications, 2019).

3. The authors seem to generally do a good job of citing relevant prior work, and mention the lack of studies look at suicide-related videos on youtube. However, I was able to a few studies that do look quite related published recently in 2020, e.g. High viewership of videos about teenage suicide on YouTube by Dagar and Falcone, and Communication about suicide in YouTube videos: Content analysis of German-language videos retrieved with method-and help-related search terms by Niederkrotenthaler et al. It may be helpful to review and mention these works. This reviewer has no connection to these works or their authors.

(Response) 

Thank you so much for the information. We have reviewed these two works and found high relevance to our manuscript. We have referred to these works and accordingly added them on the reference list.

4. Table 1 should likely contain median values, especially for the smaller groups where they may be some variation. Alternatively, the authors could consider incorporating graphics such as boxplots to describe the data. I find it a bit hard to read due to the large numbers. If continuing to use numbers, describing the numbers as the nearest thousand (e.g. 1990059 to 1990) might make the numbers easier to compare.

(Response) 

Thank you for your suggestion. In order to make the manuscript concise, as it already contains many Tables with numbers, we would like to continue using numbers. However, as suggested, we have changed the numbers as the nearest thousand to make the comparison easier for the readers. The legend provides information that 1) the unit is one thousand and 2) numbers below one thousand are written as “-”.

5. Some of the categories add up to more than 100, so I believe some categories can have multiple values. Please address in methodology if this is correct, or what happens if a category is unclear, or multi-valued e.g. a health professional who is also a survivor.

(Response) 

Thank you for this comment. 5 variables add up to more than 100 because those variables had multiple values.

Please refer to the manuscript line 442 to 446 which reads: “Among 14 variables, 5 variables including message deliverer category, whether the story is about a suicide attempt, completed suicide, or suicide ideation, illustration of method, placement of the warning sign, and placement of hotlines were multi-coded. Thus, the number of instances coded in each category may not be equal to the total number of observed instances which is 100.”

6. Thank you for addressing that you did not examine the like vs dislike ratio in your paper and it would be appropriate for further work. If you have the data readily available, I believe this could be a helpful addition to this paper as another engagement metric that may be quite different than others.

(Response) 

Thank you for this comment. We would like to leave ‘like vs dislike ratio’ metric for future work. Although ‘the number of views,’ ‘the number of likes,’ ‘the number of comments,’ and ‘like vs dislike ratio’ are all engagement metrics, they have different meanings. It would be better to explore them in the future work.

7. I would recommend a different word choice for the sentence “This study focused on one of the self-induced health concerns worldwide” in the abstract. In this context I believe it could be beneficial to describe it more directly as a result of mental health concerns, to emphasise that it is usually due to external factors rather than an individual “choice” e.g. the APA describes it as “Suicide is the act of killing yourself, most often as a result of depression or other mental illness”. Consider other choices such as “focused on a leading cause of death” or “a leading cause of death related to mental health”.

(Response) 

Thank you for your advice. As suggested, we have revised the sentence as follows: “This study focused on one of the health concerns worldwide, a leading cause of death related to mental health”.

8. Please reconsider or further elaborate on the sentences “As the deliverers of suicide-themed posts are survivors and rescuers rather than health professionals, the platform may play an important role as an arena for diagnosis. The symptoms and the reasons for suicide ideation may be explicitly stated on the platform, which may help health professionals to diagnose individuals who ideate suicide or those with suicide experience”. Has any prior work investigated this? Is there a clinical group (teens?) that posts videos about suicide often enough that this could be clinically useful? Doesn’t youtube already have a “report” button that allows something like this to happen, without the health professionals needing to view the videos directly? As a health professional, this strikes me as too far a jump without a bit more substantiation.

(Response) 

Thank you for valuable comment. One of the findings of this study showed that content delivered by survivors of suicide attempt, artists, musicians, film personnel, and one-person creators received more likes and comments. Self-expressive individuals produce videos about their experiences and thoughts where the platform makes it possible. As the platform environment enables individuals to freely upload content, some individuals share a series of stories related to their own suicide attempt. If collective coping is really happening, it will be worthwhile for the health professionals to examine how an individual overcomes suicidal thoughts and how the interaction of content uploader-content-and-viewers helps the prevention of suicide. There can be interesting dynamics of collective coping.

The ‘report’ or ‘flag’ button raises the attention of YouTube regarding inappropriate content, so that the platform can take a legitimate action, either to remove the content that violates Community Guidelines, or to put age restriction. It has little to do with diagnosing individuals. The purpose of reviewing the content may be different between the platform ‘YouTube’ and health professionals.

Thank you again for being able to read your work, and I hope you find my feedback is helpful.

(Response) Thank you for your helpful comments again. We have really enjoyed your feedback and did our best to respond to your comments. 

Reviewer #2: 

The paper targets a very interesting topic within social media. Nevertheless it requires major modifications

- The authors should provide sufficient information on the following:

(Response)

Thank you for giving us an opportunity to revise our paper. We are deeply grateful for your insightful and constructive comments. We have taken advantage of these comments in carefully preparing this revision. Added or revised parts were highlighted in the revised manuscript with track changes. Please see our detailed explanations (in blue color) in the individual responses to your comments (in black color). 

1. The language and region information of the videos analysed. Do any of the videos require age registration?

(Response) 

Thank you for this comment. The videos were analyzed in English while the researchers were located in Seoul, South Korea. The number of videos that required age registration was 5. Please refer to Table 3 in the manuscript under the category, ‘YouTube Warning.’ We have added these details in the manuscript. Please refer to line 532~534. 

2. Was the term "Suicide" searched in English? When was the search and video selection performed?

(Response) 

Thank you for this comment. The term “Suicide” was searched in English. The search and video selection were performed at one point, September 2019.

3. Were the browser cache and history cleared before each search and all filters switched off?

(Response) 

Thank you for this question. No, the browser cache and history were not cleared before the search. However, the browser was opened as Incognito Window, which enables private browsing without having to log in. All filters were switched off.

- The authors stated that no exclusion criteria were set. It will be useful to exclude unrelated contents (e.g. Music Videos, Playlists, etc.) and/or videos with a length of >10 minutes.

(Response) 

Thank you for your comment. Neither specific inclusion nor exclusion criteria was set in the sample selection process for the preliminary analysis for the purpose of extracting all possible codes relevant to suicide-themed videos. In order to build a coding system, all possible instances were included.

Videos with a length of >10 minutes and music videos were included because the authors do find all videos relevant. As the purpose of this study is to examine a general search result and as the authors have not found evidence stating people choose videos depending on the running time (length) of the video, we could not exclude those videos. We have not excluded music videos. Playlists were excluded because the unit of the analysis was an ‘individual video.’ Since playlists are discernable in the thumbnail area that shows the number of videos in the playlist, it was easily spotted and was not included in the sample.

In the final analysis, videos related to “Suicide Squad,” a superhero movie based on DC comics, were excluded because “Suicide Squad” has low relevance to the health-related issue of suicide, although the movie title does include the word ‘suicide.’

- The "Introduction" and "Literature review and research questions" sections are lengthy and contain redundant information.

(Response) 

Thank you for your advice. To respond to your advice, we have tried to eliminate some redundant information in our revised manuscript.

---

## [Decision Letter · Decision Letter 1]

24 May 2021

Suicide on YouTube: Factors engaging viewers to a selection of suicide-themed videos

PONE-D-20-36601R1

Dear Dr. Kim,

We’re pleased to inform you that your manuscript has been judged scientifically suitable for publication and will be formally accepted for publication once it meets all outstanding technical requirements.

Kind regards,

Vincenzo De Luca

Academic Editor

PLOS ONE

Additional Editor Comments (optional):

Reviewers' comments:

Reviewer's Responses to Questions

**Comments to the Author**

1. If the authors have adequately addressed your comments raised in a previous round of review and you feel that this manuscript is now acceptable for publication, you may indicate that here to bypass the “Comments to the Author” section, enter your conflict of interest statement in the “Confidential to Editor” section, and submit your "Accept" recommendation.

Reviewer #1: All comments have been addressed

2. Is the manuscript technically sound, and do the data support the conclusions?

Reviewer #1: Yes

3. Has the statistical analysis been performed appropriately and rigorously? 

Reviewer #1: Yes

4. Have the authors made all data underlying the findings in their manuscript fully available?

Reviewer #1: Yes

5. Is the manuscript presented in an intelligible fashion and written in standard English?

Reviewer #1: Yes

6. Review Comments to the Author

Reviewer #1: Thank you authors for your great work addressing my comments, I think it has added some additional rigor and reproducibility to your interesting and insightful paper. I have no additional comments.

7. PLOS authors have the option to publish the peer review history of their article (what does this mean?). If published, this will include your full peer review and any attached files.

Reviewer #1: **Yes: **John-Jose Nunez, M.D.

---

## [Editor Report · Acceptance letter]

1 Jun 2021

PONE-D-20-36601R1 

Suicide on YouTube:Factors engaging viewers to a selection of suicide-themed videos 

Dear Dr. Kim:

I'm pleased to inform you that your manuscript has been deemed suitable for publication in PLOS ONE. Congratulations! Your manuscript is now with our production department. 

Kind regards, 

on behalf of

Dr. Vincenzo De Luca 

Academic Editor

PLOS ONE